# Indoor Scene Change Captioning Based on Multimodality Data

**DOI:** 10.3390/s20174761

**Published:** 2020-08-23

**Authors:** Yue Qiu, Yutaka Satoh, Ryota Suzuki, Kenji Iwata, Hirokatsu Kataoka

**Affiliations:** 1Graduate School of Science and Technology, University of Tsukuba, Tsukuba 305-8577, Japan; yu.satou@aist.go.jp; 2National Institute of Advanced Industrial Science and Technology (AIST), Tsukuba 305-8560, Japan; ryota.suzuki@aist.go.jp (R.S.); kenji.iwata@aist.go.jp (K.I.); hirokatsu.kataoka@aist.go.jp (H.K.)

**Keywords:** image captioning, three-dimensional (3D) vision, deep learning, human-robot interaction

## Abstract

This study proposes a framework for describing a scene change using natural language text based on indoor scene observations conducted before and after a scene change. The recognition of scene changes plays an essential role in a variety of real-world applications, such as scene anomaly detection. Most scene understanding research has focused on static scenes. Most existing scene change captioning methods detect scene changes from single-view RGB images, neglecting the underlying three-dimensional structures. Previous three-dimensional scene change captioning methods use simulated scenes consisting of geometry primitives, making it unsuitable for real-world applications. To solve these problems, we automatically generated large-scale indoor scene change caption datasets. We propose an end-to-end framework for describing scene changes from various input modalities, namely, RGB images, depth images, and point cloud data, which are available in most robot applications. We conducted experiments with various input modalities and models and evaluated model performance using datasets with various levels of complexity. Experimental results show that the models that combine RGB images and point cloud data as input achieve high performance in sentence generation and caption correctness and are robust for change type understanding for datasets with high complexity. The developed datasets and models contribute to the study of indoor scene change understanding.

## 1. Introduction

There have been significant improvements in artificial intelligence (AI) technologies for human–robot interaction (HRI) applications. For example, modern intelligent assistants (e.g., Google Assistant [1]) enable the control of household appliances through speech and allow remote home monitoring. HRI experiences can be improved through the use of AI technologies, such as the semantic and geometric understanding of 3D surroundings [2,3,4,5], the recognition of human gestures [6,7], actions [8,9], emotions [10,11], speech recognition [12,13], and dialog management [14,15]. A fundamental problem in indoor scene understanding is that scenes often change due to human activities, such as the rearranging of furniture and cleaning. Therefore, understanding indoor scene changes is essential for many HRI applications.

Developments in graphic processing units and convolutional neural network (CNN)-based methods have led to tremendous progress in 3D recognition-related studies. Various 2D approaches have been adapted for 3D data, such as recognition [3,4,5], detection [16], and segmentation [17]. Researchers have proposed a series of embodied AI tasks that define an indoor scene and an agent that explores the scene and answers vision-related questions (e.g., embodied question answering [18,19]), or navigates based on a given instruction (e.g., vision-language navigation [20,21]). However, most 3D recognition-related studies have focused on static scenes. Scene change understanding is less often discussed despite its importance in real-world applications.

Vision and language tasks, including visual question answering [22,23,24], image captioning [25,26,27,28,29,30], and visual dialog [31,32], have received much attention due to their practicality in HRI applications. These tasks correlate visual information with language. The image captioning task aims to describe image information using text, and thus can be used to report scene states to human operators in HRI applications. Several recent image captioning methods describe a scene change based on two images of the scene [33,34]. However, they use single-view image inputs and neglect the geometric information of a scene, which limits their capability in scenes that contain occlusions. Qiu et al. [35] proposed scene change captioning based on multiview observations made before and after a scene change. However, they considered only simulated table scenes with limited visual complexity.

To solve the above problems, we propose models that use multimodality data of indoor scenes, including RGB (red, green, and blue) images, depth images, and point cloud data (PCD), which can be obtained using RGB-D (RGB-Depth) sensors, such as Microsoft Kinect [36], as shown in Figure 1. We automatically generated large-scale indoor scene change caption datasets that contain observations made before and after scene changes in the form of RGB and depth images taken from multiple viewpoints and PCD along with related change captions. These datasets were generated by sampling scenes from a large-scale indoor scene dataset and objects from two object model datasets. We created scene changes by randomly selecting, placing, and rearranging object models in the scenes. Change captions were automatically generated based on the recorded scene information and a set of predefined grammatical structures.

We also propose a unified end-to-end framework that generates scene change captions from observations made before and after a scene change, including RGB images, depth images, and PCD. We conducted extensive experiments on input modalities, encoders, and ensembles of modalities with datasets under various levels of complexity. Experimental results show that the models that combine RGB images and PCD can generate change captions with high scores in terms of conventional image caption evaluation metrics and high correctness in describing detailed change information, including change types and object attributes. The contributions of our work are four-fold:We automatically generated the first large-scale indoor scene change caption dataset, which will facilitate further studies on scene change understanding.We developed a unified end-to-end framework that can generate change captions from multimodality input, including multiview RGB images, depth images, and PCD, which are available in most HRI applications.We conducted extensive experiments on various types of input data and their ensembles. The experimental results show that both RGB images and PCD are critical for obtaining high performance, and that the use of PCD improves change type prediction robustness. These results provide perspectives for enhancing performance for further research.We conducted experiments using datasets with various levels of complexity. The experimental results show that our datasets remain challenging and can be used as benchmarks for further exploration.

## 2. Related Work

### 2.1. 3D Scene Understanding

CNN-based methods have promising performance in various 3D scene understanding tasks, such as 3D object recognition [3,4,5], 3D detection [16], 3D semantic segmentation [17], and shape completion [37]. These methods use CNN structures to learn the underlying 3D structures based on data in various formats, such as multiview RGB images, RGB-D images, PCD, and meshes.

Su et al. [38] proposed a network for 3D object recognition based on multiview RGB images. They proposed a multiview CNN (MVCNN) structure for aggregating information via a view pooling operation (max or average pooling) from the CNN features of multiview images. Kanezaki et al. [39] proposed a framework for feature extraction from multiview images that predicts object poses and classes simultaneously to improve performance. Esteves et al. [40] suggested that existing MVCNNs discard useful global information and proposed a group convolutional structure to better extract the global information contained in multiview images. Eslami et al. [2] proposed a generative query network (GQN) that learns 3D-aware scene representations from multiview images via an autoencoder structure. Several studies have focused on 3D understanding based on RGB-D data. Zhang et al. [41] proposed a network for depth completion from a single RGB-D image that predicts pixel-level geometry information. Qi et al. [42] proposed a 3D detection framework that detects objects in RGB images and fuses depth information to compute 3D object regions. Recent CNNs that utilize PCD have also shown promising results. Qi et al. proposed PointNet [3], which is a structure for extracting features from raw PCD via the aggregation of local information by symmetric functions (e.g., global max pooling). They later proposed PointNet++ [4] for obtaining better local information. Zhang et al. [5] proposed a simple yet effective structure that aggregates local information of PCD via a region-aware max pooling operation.

Considering the availability of RGB-D data in HRI applications, we propose models that use multiview RGB and depth images and PCD. We adopt an MVCNN and a GQN for scene understanding based on RGB images, an MVCNN for aggregating multiview depth information, and PointNet for processing PCD.

### 2.2. Indoor Scene Datasets

Due to the high complexity and diversity of visual information, training a CNN-based indoor scene understanding method usually requires a massive amount of data. SUNCG [43] is a widely used dataset that consists of simulated 3D indoor scenes generated using computer graphics technologies. Several indoor scene datasets with scanned models of real scenes have recently been made publicly available [44,45,46]. The Gibson [44] dataset consists of 572 scenes and a simulator, which allows training for multiple embodied AI tasks, such as visual navigation. Matterport3D [45] contains 90 indoor scenes that are densely annotated with semantic labels. The Replica [46] dataset consists of 18 high-resolution (nearly photorealistic) scenes.

Several datasets for embodied AI tasks have been built based on the above 3D datasets. The Embodied Question Answering (EQA) v1.0 [18] dataset consists of scenes sampled from the SUNCG dataset with additional question–answer pairs. The authors further extended the EQA task for realistic scene setting by adapting the Matterport3D dataset to their Matterport3D EQA dataset [19]. The Room-to-Room dataset [20] added navigation instruction annotation to the Matterport3D dataset for the vision-language navigation task. In these datasets, the states of the scenes are static. Qiu et al. [35] proposed a simulated dataset for scene change captioning from multiview RGB images. However, they generated scenes with a solid background color, limiting the visual complexity. In contrast, we combine the Matterport3D dataset with two open source object model datasets, namely, NEDO item database [47] and YCB dataset [48], for creating scene change datasets, where scene changes are constructed by rearranging objects in 3D scenes. To the best of our knowledge, our dataset is the first large-scale indoor scene change dataset.

### 2.3. Change Detection

Change detection is a long-standing task in computer vision due to its practicality in real-world applications, such as scene anomaly detection, and disaster influence analysis. Change detection from street view images or videos has attracted much attention because it allows algorithms to focus on changed regions, decreasing the cost of image or video recognition [49,50]. Alcantarilla et al. [49] proposed a method that first reconstructs 3D geometry from video input and then inputs coarsely registered image pairs into a deconvolutional network for change detection. Zhao et al. [50] proposed a method with an encoder–decoder structure for pixel-level change detection based on street view images.

Change detection is also important in robot applications [51,52,53]. Ambrus et al. [51] proposed a method that distinguishes static and dynamic objects by reconstructing and comparing PCD of a room scene observed at different time steps. Fehr et al. [52] proposed a 3D reconstruction method that reconstructs static 3D scenes based on RGB-D images of scenes with dynamic objects. Jinno et al. [53] proposed a framework for updating a 3D map by comparing the existing 3D map with newly observed 3D data that may contain new or removed objects.

Existing change detection methods that utilize RGB images lack 3D geometry understanding. Several methods have been proposed for detecting changes from 3D data in various formats, such as RGB-D images and PCD, for robot applications. However, most works are limited to relatively small-scale datasets and do not specify detailed changes, such as the attributes of changed objects. In contrast, we consider change detection based on multimodality input, including RGB and depth images and PCD. Our models describe detailed scene changes, including change types and object attributes.

### 2.4. Change Captioning

The image captioning task has been widely discussed. Various image captioning methods have been proposed to achieve high-performance sentence construction by using attention mechanisms [25,26] or exploring relationships between vision and language [27,28]. Generating image captions with high diversity has also been widely discussed [29,30]. However, most existing image captioning methods generate descriptions from single-view images.

Several recent works have discussed captioning based on images that include scene changes [33,34,35]. Difference Description with Latent Alignment (DDLA) [33] generates change descriptions from two video frames observed from different time steps of a given scene. In DDLA, an image indicating the pixel-level difference between input frames is computed and a CNN is used for generating captions from this difference image. The DUal Dynamic Attention model (DUDA) [34] uses a dual attention structure for focusing regions of images before and after a change and a dynamic attention structure, which dynamically selects information from image features before or after a change, or the difference between them. DUDA is more robust to camera transformation compared to DDLA. However, both DDLA and DUDA neglect the 3D geometry information of scenes and thus are less suitable for scenes with occlusions. Qiu et al. [35] proposed a method that generates a compact scene representation from multiview images and then generates captions based on the scene representation. However, they performed experiments using scenes with solid colored backgrounds, and only considered RGB images from a fixed number of cameras.

In contrast, we explore and evaluate several input modalities, namely, RGB and depth images (with random camera position changes), and PCD. We conducted extensive experiments on various ensembles of these modalities. We also conducted experiments on datasets with complex and diverse visual information.

## 3. Approach

In robot applications, the ability to recognize scene changes is essential. We propose a framework that generates scene change captions from image pairs taken before and after a scene change. Our framework correlates the after-change scene with the before-change scene and provides detailed change descriptions, including change types and object attributes. Due to the availability of RGB-D data in robot applications, we developed models that use RGB and depth images of scenes observed from multiple viewpoints and PCD. Our framework can be trained end-to-end using raw images, PCD, and related change captions. Moreover, our framework enables inputs from one or more modalities. In the following subsections, we give the details of the proposed framework.

### 3.1. Overall Framework

As shown in Figure 2, our framework generates a change caption from the input of observations taken before and after a scene change. A scene observation consists of RGB and depth images observed from multiple viewpoints and PCD.

Our framework comprises three components: a scene encoder, which processes input modalities with respective encoders; a scene fusion component, which combines the features of observations taken before and after a scene change; and a caption generator, which generates text captions from fused scene representations. Our framework can be further enhanced by adding more modalities, such as normal maps. Moreover, the scene fusion component and caption generator can also be improved by adopting novel approaches. We give the details of these three components in the remaining subsections.

### 3.2. Scene Encoder

The scene encoder component transforms an observation of a scene consisting of multiview RGB, depth images, and PCD into a feature vector to express semantic and geometric information for the scene. As shown in Figure 2, we first extract feature vectors from multiview RGB images, multiview depth images, and PCD separately with respective encoders. Then, the information is aggregated via a concatenation operation. We experimented with two encoder structures, namely, MVCNN and GQN, for encoding multiview RGB images and MVCNN for depth images.

RGB/depth encoder (MVCNN): This network is adapted from Su et al. [38]. In our implementation (Figure 3a), we first extract features from each viewpoint (we transformed depth images to RGB images through the applyColorMap function with mapping parameter COLORMAP_JET defined in OpenCV [54]) via ResNet101 [55]. Then, we apply convolution operation (separated weights) to extracted features and compute a weight vector via fully connection and softmax function. Then, we use another convolution layer to ResNet101-extracted features and multiply the output with the weight vector. Finally, a 4×4×128×8-dimensional feature vector is obtained.

RGB encoder (GQN): Eslami et al. [2] proposed GQN, which recovers an image from a given viewpoint through a scene representation network and a generation network. We adapt GQN (tower-structure [2], Figure 3b) to extract a scene representation from multiview RGB images in two stages. During the pretraining stage, we train the overall GQN using multiview scene images. Then, we discard the generation network of GQN and use the pretrained scene representation network to aggregate information from multiview images.

PCD encoder (PointNet): We use PointNet, proposed by Qi et al. [3], for extracting features from PCD. PointNet transforms raw PCD into a feature vector that can be used for a range of downstream tasks, such as classification and segmentation. The detailed structure used in this work is shown in Figure 3c.

After processing these modalities separately, we resize features to 1×1×k-dimensional vectors (*k* is different for three modalities) and combine features using a concatenation operation. The concatenation operation makes it possible to change the number of input feature vectors, enabling both single- and multimodality inputs.

### 3.3. Before- and After-Change Scene Fusion

We process the observation pairs taken before and after a scene change using the process described in the previous subsection. The two feature vectors are combined via the fusion method proposed by Park et al. [34].

Specifically, we denote the feature vector of the before-change scene as $r_{\mathrm{bef}}$ and that of the after-change scene as $r_{\mathrm{aft}}$. We first compute the vector difference $r_{\mathrm{diff}}$ of $r_{\mathrm{bef}}$ and $r_{\mathrm{aft}}$ via the following formulation.
(1)$$r_{\mathrm{diff}}=r_{\mathrm{aft}}-r_{\mathrm{bef}}\;$$

Then, we concatenate $r_{\mathrm{bef}}$, $r_{\mathrm{aft}}$, and $r_{\mathrm{diff}}$ to create a compact feature vector as the caption generator’s input.

### 3.4. Caption Generator

The caption generator predicts a change caption from the output of the scene fusion component. As shown in Figure 2, we used a two-layer long short-term memory (LSTM) [56] structure for caption generation. Notably, the caption generator can be replaced by other language models, such as a transformer [57].

The overall network is trained end-to-end with the following loss function to minimize the distance *L* between generated captions x and ground truth captions y:(2)L=−log(P(y|x))

## 4. Indoor Scene Change Captioning Dataset

Due to the high semantic and geometric complexity, training tasks targeting indoor scene understanding require a large amount of training data. Moreover, the construction of indoor scene datasets often requires a lot of manual labor. To the best of our knowledge, there is no existing large-scale indoor scene change captioning dataset.

To solve the above problems, we propose an automatic dataset generation process for indoor scene change captioning based on the existing large-scale indoor scene dataset Matterport3D [45] and two object model datasets, namely, NEDO [47] and YCB [48]. We create the before- and after-change scenes by arranging object models in indoor scenes. We set four atomic change types: add (add an object to a scene), delete (delete an object), move (change the position of an object), and replace (replace an object with a different one). We also set a distractor type in our dataset that indicates only camera position changes compared to original scenes. Implementing changes in the original 3D dataset is an alternative approach for creating datasets; however, this will result in artifacts, such as large holes after an object is deleted or moved from its original position. Thus, we sample object models from existing object model datasets and arrange them in 3D scenes to create before- and after-change scenes.

In the following subsections, we give the details of the automatic dataset generation process and the datasets created for experiments.

### 4.1. Automatic Dataset Generation

#### 4.1.1. Scene and Object Models

We generated before- and after-change scene observation pairs based on arranging object models in 3D scenes. We used the Matterport3D dataset (consisting of 3D mesh models of 90 buildings with 2056 rooms) as our scene source. We selected 115 cuboid rooms that contain fewer artifacts (e.g., large holes in geometry) from the Matterport3D dataset. The object models used in our dataset generation were sampled from the NEDO and YCB datasets. We list the object class and instances in Table 1.

#### 4.1.2. Virtual Camera Setups

We took RGB and depth images from multiple camera viewpoints along with PCD for each scene observation. To obtain an overall observation for each room scene, as shown in Figure 4, we used cuboid rooms and set eight virtual cameras (four corners and four centers of edges of the ceiling) for observing scenes. Each virtual camera was set to look at the center of the room. In addition, to enhance robustness to camera position transformation, we added random offsets of [–10.0 cm, +10.0 cm] in three dimensions for each camera during the dataset acquisition.

The data acquisition process can be implemented by using a single RGBD camera to observe a scene multiple times from various camera viewpoints.

#### 4.1.3. Generation Process

We use AI Habitat [58] as the simulator for data acquisition. AI habitat enables generating RGB and depth images of given viewpoints from a mesh file. We generated each before- and after-change scene observation pair and related change captions in four steps. We first randomly selected a room scene from the scene sets and object models (three to five) from the object sets. The AI Habitat simulator provides a function named “get_random_navigable_point()”, which computes a random position where the agent can walk on based on the mesh data and semantic information (semantic label information, such as “floor” and “wall”, for each triangle vertex, provided in Matterport3D dataset). In the second step, we utilized the function to obtain random navigable positions and arranged objects on those positions. We took eight RGB and depth images and generated PCD as the original scene observation through the AI Habitat simulator. The Matterport3D dataset provides mesh data of every building and position annotation for each room. We generated PCD by transforming vertices of mesh data (triangular mesh) into points of PCD. We extracted PCD for each room from PCD of building based on room position annotation (3D bounding box annotation of rooms provided by the Matterport3D dataset). Next, we implemented the four change types (add, delete, move, and replace) for the original scene along with a distractor (only camera position transformation) and obtained scene observations. The change information, including change type and object attributes, was recorded. Finally, we generated five change captions for each change type and the distractor based on the recorded change information and predefined sentence structure templates (25 captions in total for each scene). We show an example of our dataset in Figure 5. The above process makes it easy to generate datasets with various levels of complexity by adjusting scene and object numbers, change types, and sentence templates.

We currently used PCD generated from meshes, which contains fewer artifacts, such as holes and less occlusion. To further improve the practicality of our method, we plan to use PCD generated from RGBD images and conduct experiments to discuss the effects of occlusion and artifacts.

### 4.2. Dataset Statistics

We generated dataset s15_o10 with 9000 scenes for training and 3000 scenes for testing. In s15_o10, we used 15 scenes and 10 object models (10-object set-up in Table 1). We used the s15_o10 dataset to evaluate the performance obtained with various input modalities, encoders, and ensembles.

To evaluate model performance under more complex scene settings, we adjusted the number of scenes and objects and generated dataset s15_o85 with 85 object models (85-object set-up in Table 1) and dataset s100_o10 with 100 scenes. The other settings of s15_o85 and s100_o10 are the same as those for s15_o10. The detailed dataset statistics are shown in Table 2. Experiments with these three datasets are presented below.

## 5. Experiments

We used datasets s15_o10, s15_o85, and s100_o10 for training and evaluation. Specifically, we first used s15_o10 for the comparison of different input modalities, encoders (MVCNN and GQN for RGB images), and ensembles of input modalities. We then used s15_o85 and s100_o10 for assessing the models’ abilities under more complex scene setups with an increased number of objects and scenes.

We adopted several conventional image captioning evaluation metrics in each experiment. In addition to these metrics, we conducted a caption correctness evaluation to examine the detailed information given by the generated captions (change types and object attributes).

### 5.1. Evaluation Metrics

We used four conventional evaluation metrics widely adopted in image captioning: BLEU-4 [59], ROUGE [60], SPICE [61], and METEOR [62]. These metrics evaluate the similarities between the generated captions and the ground truth captions. BLEU-4 is used to evaluate the recall of words or phrases (multiple words) of generated captions in the ground truth captions. ROUGE evaluates the recall of ground truth captions in generated captions. SPICE considers the correctness of the sentence structures of generated captions. METEOR introduces the similarity between words to encourage the generation of captions with diverse words.

The correctness of change type and object attributes is important in the change captioning task. Therefore, in addition to the above metrics, we conducted a caption correctness evaluation. We neglect the correctness of the sentence structure and extract change type, class, color, and object (including class and color, such as “red cup”) from the generated captions and compute the accuracy when compared to the ground truth captions. This evaluation indicates how well the generated captions reflect the detailed change information.

### 5.2. Implementation Details

Here, we give the details of all the implementations. We set the input image size of both MVCNN and GQN to 64×64. We set the point number of PCD to 5000 for PointNet by random selecting points from PCD of rooms. For the pretraining process of GQN, we set the learning rate to $10^{-4}$ and trained the overall GQN network for 10 epochs in all experiments. For the overall framework training (including all single modalities and ensembles), we set the learning rate to $10^{-3}$ for PointNet and $10^{-4}$ for MVCNN and the decoder. All ablations were trained for 40 epochs. We used the Adam optimizer in all experiments.

### 5.3. Experiments on Input Modality and Model Ablations

We first used dataset s15_o10 to evaluate the performance of various input modalities, encoders, and ensembles. Here, we implemented four single-modality ablations, namely, depth images with the MVCNN encoder, RGB images with MVCNN and GQN encoders, and PCD with the PointNet encoder. We implemented five two-modality ablations containing two different input modalities, where the RGB images were processed using MVCNN and GQN encoders. We also implemented two ensembles with three modalities, where different RGB encoders were adopted.

Evaluation results on the test split of s15_o10 in terms of conventional evaluation metrics are shown in the top 11 rows of Table 3. The four single-modality ablation results show that PCD with the PointNet encoder obtained the best performance and that depth with the MVCNN encoder obtained the lowest scores for all metrics. RGB images with the GQN encoder outperformed the ablation with the MVCNN encoder. PCD contain geometric and object edge information, which is an advantage in a task that requires recognizing object change and detailed object attributes. GQN was trained to obtain a compact scene representation from multiview images, which likely made it better at correlating multiview information compared to MVCNN. Depth images do not contain color information and it is difficult to obtain object shapes from them, making it challenging to understand a scene change from only depth images. We found that although depth images alone performed poorly, ensembles containing RGB (MVCNN or GQN encoder) and depth images outperformed RGB images alone. We think that this resulted from the geometric information in depth images, which is difficult to extract from RGB images. The two ensembles with three input modalities outperformed all single-modalities and ensembles with two input modalities composed of their subsets. Scene change captioning performance can thus be enhanced by using both geometric and RGB information.

The caption correctness evaluation results are shown in the top 11 rows in Table 4. We found that ablation with RGB input (GQN encoder) outperformed PCD input in terms of object correctness, whereas PCD obtained higher accuracy in change type prediction (single modality). The abundant geometric and edge information contained in PCD is beneficial for change type prediction. We also found that models with both RGB and PCD (or depth) input obtained higher object correctness than that of single modalities. This result indicates that combining geometric and RGB information leads to a better understanding of detailed object information, which is critical for obtaining high performance in this task.

We show two example results in Figure 6. For the first example (object moved), all ensembles predict change captions correctly. Ablations with depth images and RGB images (MVCNN encoder) correctly determined the related object attributes (cyan minicar) but failed to predict the correct change type (move). In contrast, the single-modality ablation with PCD predicted the correct change type but gave the wrong color. For the second example (distractor with no object change), all models with PCD and the ensemble with RGB (GQN encoder) and depth images gave correct change captions. However, the other models predicted wrong change captions. These results indicate that combining modalities enhances performance, and that PCD can provide geometry information, which is beneficial for predicting change type.

### 5.4. Experiments on Dataset Complexity (Object Class Number)

In this experiment, we evaluated model performance under a more complex set-up s15_o85, where the object instance number was 85 (10 objects in s15_o10). We conducted experiments on four single-modality and two three-modality ablations.

The results of conventional evaluation metrics are shown in the middle six rows of Table 3. The performance for all ablations is lower than that for s15_o10. Notably, the performance of models with RGB images and the GQN encoder degraded significantly. This result indicates that the GQN encoder is less robust to scene set-ups with high complexity. Similar to s15_o10, for single-modality input, PCD with PointNet performed the best. Ensembles outperformed single modalities.

The caption correctness evaluation is shown in the middle six rows of Table 4. The performance of all modalities degraded compared to that for s15_o10 in terms of object correctness. However, models with PCD, including both single modalities and ensembles, tended to be more robust for change type prediction. This result indicates that the geometry and edge information make change type prediction more consistent.

We show one example result (object deleted) in Figure 7 (top). All single modalities failed to give correct captions, whereas the two ensembles predicted the correct caption. Single-modality models with depth or RGB images gave the wrong change type, whereas that with PCD correctly predicted the change type (delete). s15_o85 is more challenging than s15_o10 because it included more objects (85 vs. 10). Combining different modalities is effective for handling datasets with relatively high complexity.

### 5.5. Experiments on Dataset Complexity (Scene Number)

Here, we evaluate four single-modality and two three-modality ablations with dataset s100_o10, which included 100 scenes (15 scenes in s15_o10 and s15_o85).

The experimental results (bottom six rows of Table 3) for conventional evaluation metrics show that the performance of all ablations degraded compared to that for s15_o10. This is especially true for RGB images with the GQN encoder, which indicates that GQN is less suitable for large-scale scene datasets. Similar to s15_o10, PCD with PointNet obtained the highest scores and depth images showed poor performance. Ensembles tended to outperform single modalities. The caption correctness results (bottom six rows of Table 4) also show that performance degraded (especially for object correctness) compared to that for s15_o10. PCD with the PointNet encoder and ensembles tended to be more robust for caption type prediction.

One example result (object addition) for s100_o10 is shown in Figure 7 (bottom). Here, PCD with the PointNet encoder and the ensemble model with PCD, depth, and RGB images (MVCNN encoder) correctly predicted the change caption. For s100_o10, which contains 100 scenes, the performance of GQN encoder dramatically degraded, which may have influenced the performance of the ensemble model.

### 5.6. Discussion

The experiments using s15_o10, s15_o85, and s100_o10 indicate that for single modalities, PCD with the PointNet encoder consistently obtained the highest scores for most conventional caption evaluation metrics and the depth images with MVCNN encoder obtained the low scores in most experiments. A further evaluation of caption correctness indicated that models with RGB images (both MVCNN and GQN encoders) performed well in recognizing object attributes and those with PCD performed well in predicting change types. Model performance can be enhanced by adopting ensembles. In addition, both the RGB images and PCD are crucial for obtaining high performance with ensembles. Additional depth images improve the performance of models with RGB images but degrade that of models with PCD alone.

We found that for all modalities, performance degraded in experiments using datasets with more objects (s15_o85) and more scenes (s100_o10). However, models with PCD and the PointNet encoder tended to be relatively robust for change type prediction. Regarding the two types of RGB encoder, GQN outperformed MVCNN for s15_o10. For s15_o85 and s100_o10, the performance of GQN encoder significantly degraded, becoming worse than that of MVCNN. This result indicates that compared to MVCNN, the GQN network is less suitable for in large-scale scene-setting.

The experimental results reported here will facilitate future research on scene change understanding and captioning. To understand and describe scene changes, both geometry and color information are critical. Because we used the concatenation operation to aggregate the information of various modalities, introducing an attention mechanism to dynamically determine the needed features could help enhance model performance. We evaluated model performance under complex scene settings through experiments using s15_o85 and s100_o10. It is important to further study the adaptiveness of models to scene complexity by conducting more experiments using diverse dataset setups. PCD used in this study consist of scenes and object models with integral shapes that are beneficial in change captioning task. However, in real-world applications, obtaining PCD with integral object shapes is challenging. One way to enhance the practicality of our work is to conduct experiments on partially observed PCD further.

To the best of our knowledge, our work is the first attempt for indoor scene change captioning. It is an essential future direction to adapt existing indoor scene change detection methods, such as those in [51,52,53], to the change detection task and conduct comparison experiments between our work and existing change detection methods.

## 6. Conclusions

This study proposes an end-to-end framework for describing scene change based on before- and after-change scenes observed by multiple modalities, including multiview RGB and depth images and PCD. Because indoor scenes are constantly changing due to human activities, the ability to automatically understand scene changes is crucial for HRI applications. Previous scene change detection methods do not specify detailed scene changes, such as change types or attributes of changed objects. Existing scene change captioning methods use RGB images and conduct experiments using small-scale datasets with limited visual complexity. We automatically generated large-scale indoor scene change captioning datasets with high visual complexity and proposed a unified framework that handles multiple input modalities. For all experiments, models with PCD input obtained the best performance among single-modality models, which indicates that the geometry information contained in PCD is beneficial for change understanding. The experimental results show that both geometry and color information are critical for better understanding and describing scene changes. Models with the RGB images and PCD have promising performance in scene change captioning and exhibit high robustness for change type prediction. Because we used a concatenation operation for aggregating information from various modalities, model performance could be enhanced by introducing an attention mechanism to determine the required features. Experiments on datasets with high levels of complexity show that there is still room for improvement, especially for object attribute understanding. We plan to conduct more experiments on the adaptiveness to scene complexity.

## Figures and Tables

**Figure 1 sensors-20-04761-f001:**
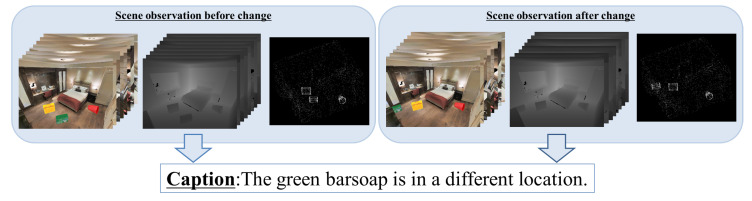
Illustration of indoor scene change captioning from multimodality data. From the input of two observations (consisting of RGB (red, green, and blue) and depth images captured by multiple virtual cameras and point cloud data (PCD)) of a scene observed before and after a change, the proposed approach predicts a text caption describing the change. The multiple RGB and depth images are obtained from multiple viewpoints of the same scene via virtual cameras. Each virtual camera takes an RGB and a depth image from a given viewpoint.

**Figure 2 sensors-20-04761-f002:**
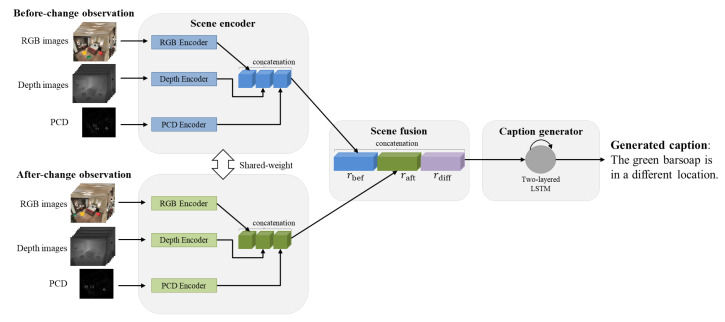
Overall framework. From before- and after-change scene observations of a scene, the proposed framework generates a text caption describing the scene change via three components, namely, a scene encoder that encodes the input composed of various modalities, a scene fusion component that combines the before- and after-change features $r_{\mathrm{bef}}$ and $r_{\mathrm{aft}}$, a caption generator that generates a caption from the output of the scene fusion component.

**Figure 3 sensors-20-04761-f003:**
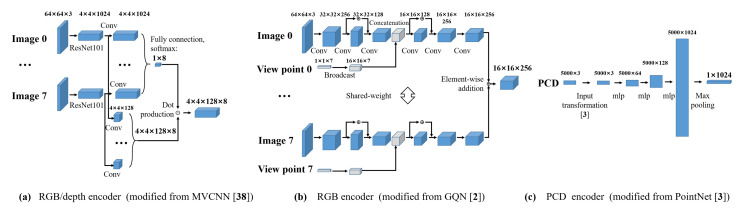
Detailed network structure for the encoders used in this study.

**Figure 4 sensors-20-04761-f004:**
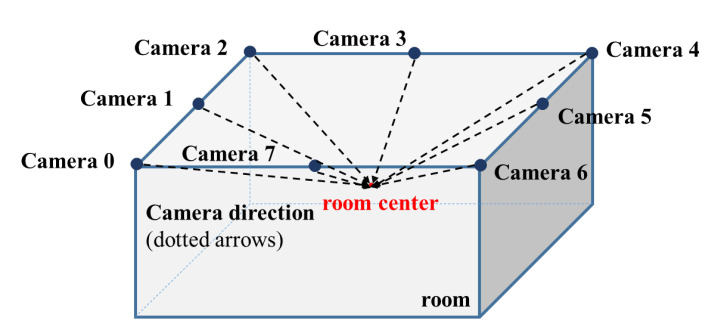
Virtual camera set-ups. Eight virtual cameras (four corners and four centers of edges of the ceiling) are set to look at the center of the room.

**Figure 5 sensors-20-04761-f005:**
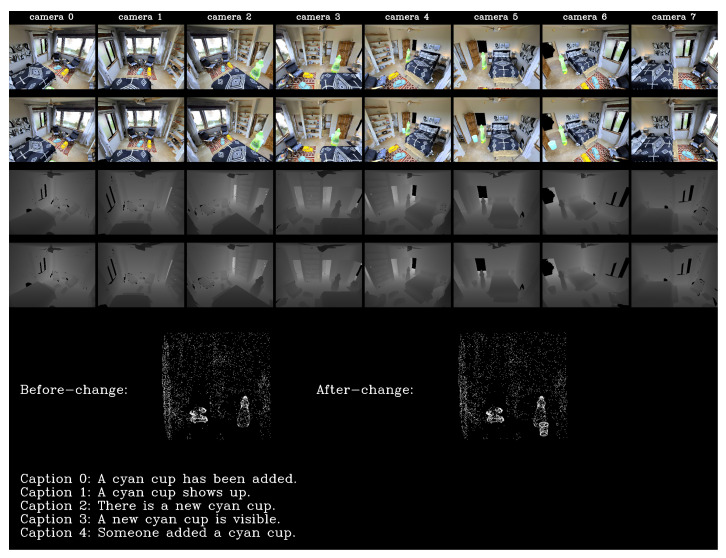
Dataset instance example of adding an object. From the top row: before-change RGB images observed from eight virtual cameras; after-change RGB images; before-change depth images; after-change depth images; before- and after-change PCD; five ground truth change captions.

**Figure 6 sensors-20-04761-f006:**
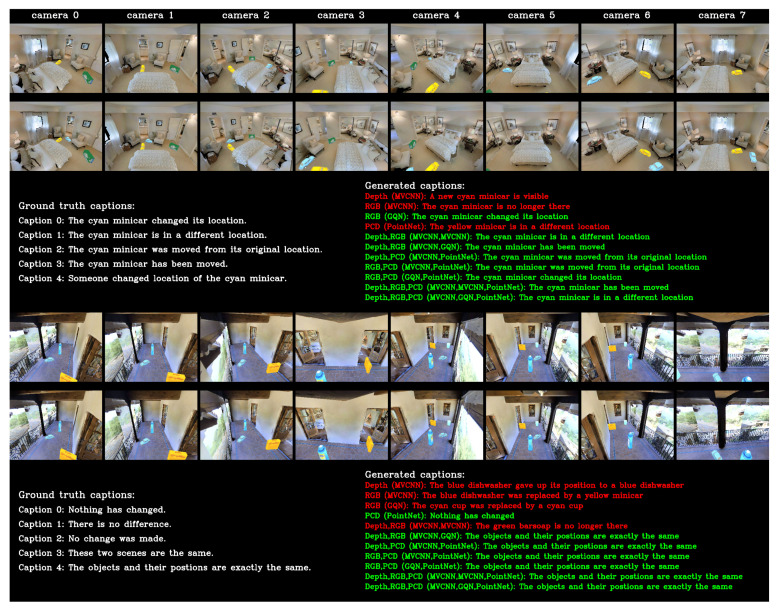
Example results for s15_o10. From the top row: before-change RGB images observed from eight virtual cameras for first example; after-change RGB images; ground truth and generated captions for various models; before-change RGB images for second example; after-change RGB images (distractor with only camera position transformation); ground truth and generated captions. Correct captions are shown in green and false captions are shown in red.

**Figure 7 sensors-20-04761-f007:**
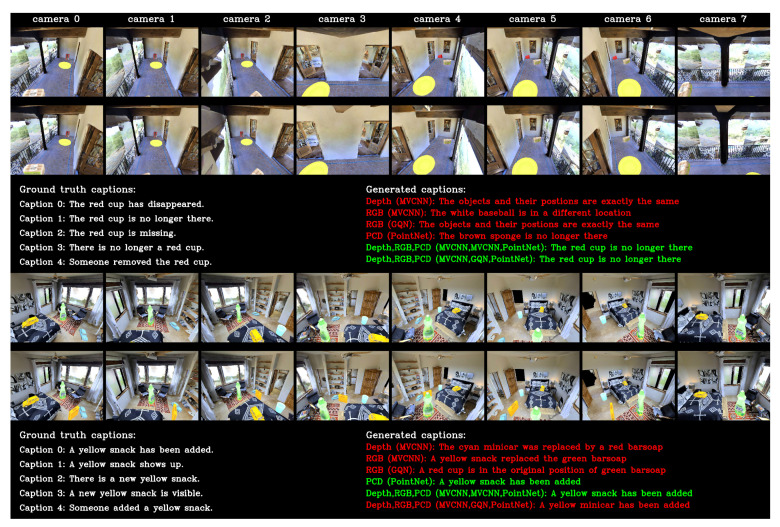
Example results for s15_o85 and s100_o10. From the top row: before-change RGB images observed from eight virtual cameras for example from s15_o85; after-change RGB images; ground truth and generated captions for various models; before-change RGB images for example from s100_o10; after-change RGB images; ground truth and generated captions. Correct captions are shown in green and false captions are shown in red.

**Table 1 sensors-20-04761-t001:** Set-ups of object classes and instances for datasets used in this study.

Setups	Class (Number of Instances)
10-object setup	barsoap (2); cup (2); dishwasher (2); minicar (2); snack (2)
85-object setup	snack (12); dishwasher (11); tv dinner (11); minicar (7); barsoap (5); cup (5); plate (3); soft drink (3); sponge (3); air brush (2); baseball (2); bowl (2); facial tissue (2); magic marker (2); sauce (2); water bottle (2); weight (2); glass (1); glue (1); shampoo (1); soccer ball (1); tape (1); teddy bear (1); timer (1); toilet tissue (1); tooth paste (1)

**Table 2 sensors-20-04761-t002:** Statistics for datasets used in this study.

Dataset	No. of Scenes	No. of Captions	Change Types	Viewpoints	Scene Types	Object Classes	Objects Per Scene
	(Train/Test)	(Train/Test)					
s15_o10	(9000/3000)	(225,000/75,000)	5	8	15	10	2–6
s15_o85	(9000/3000)	(225,000/75,000)	5	8	15	85	2–6
s100_o10	(9000/3000)	(225,000/75,000)	5	8	100	10	2–6

**Table 3 sensors-20-04761-t003:** Results of evaluation using s15_o10 (top 11 rows), s15_o85 (middle 6 rows), and s100_o10 (bottom 6 rows). The highest scores are shown in bold.

Modality (Encoder)	ROUGE	SPICE	METEOR	BLEU-4 [59]					
	[60]	[61]	[62]	Overall	Add	Delete	Move	Replace	Distractor
Depth (MVCNN)	58.36	18.46	26.50	36.93	21.30	39.15	42.22	32.90	45.11
RGB (MVCNN)	65.53	26.52	34.80	49.43	25.73	49.83	54.18	51.83	60.60
RGB (GQN)	81.98	36.02	48.49	71.01	59.59	77.31	59.52	68.52	85.80
PCD (PointNet)	89.75	35.32	49.34	76.17	62.89	79.19	77.98	71.73	96.19
Depth,RGB (MVCNN,MVCNN)	80.74	34.44	46.30	67.58	55.04	71.82	69.25	63.41	75.80
Depth,RGB (MVCNN,GQN)	84.52	36.51	50.71	74.46	63.99	77.12	65.49	71.48	90.22
Depth,PCD (MVCNN,PointNet)	84.41	27.74	43.03	65.91	54.84	68.73	59.37	58.00	98.85
RGB,PCD (MVCNN,PointNet)	89.31	36.40	50.50	75.65	65.97	80.02	75.76	70.93	91.00
RGB,PCD (GQN,PointNet)	92.36	41.36	57.00	84.74	81.65	**89.87**	70.67	80.77	**99.41**
Depth,RGB,PCD (MVCNN,MVCNN,PointNet)	89.87	35.99	52.11	79.02	64.18	82.38	77.06	73.49	96.51
Depth,RGB,PCD (MVCNN,GQN,PointNet)	**93.38**	**44.29**	**58.46**	**86.33**	**83.32**	89.69	**80.21**	**82.84**	98.55
Depth (MVCNN)	57.56	15.27	24.19	36.46	12.44	33.70	30.21	19.83	74.73
RGB (MVCNN)	57.31	15.17	24.80	35.67	18.08	31.78	32.46	29.78	59.91
RGB (GQN)	63.90	18.94	29.05	43.75	23.83	42.51	34.60	34.96	77.00
PCD (PointNet)	80.44	21.04	34.98	52.02	33.26	52.79	**55.59**	53.63	52.66
Depth,RGB,PCD (MVCNN,MVCNN,PointNet)	**81.26**	23.76	**38.72**	**59.79**	**43.52**	**60.90**	43.64	**55.96**	95.98
Depth,RGB,PCD (MVCNN,GQN,PointNet)	78.83	**24.49**	36.86	57.11	39.74	59.96	41.75	43.27	**97.69**
Depth (MVCNN)	56.06	16.51	24.56	33.46	16.45	31.00	35.64	29.72	48.53
RGB (MVCNN)	65.72	25.02	33.80	47.01	31.74	43.49	51.31	51.69	50.32
RGB (GQN)	65.23	23.91	32.24	47.10	35.31	46.79	47.36	39.92	58.85
PCD (PointNet)	82.93	26.29	41.48	63.30	47.22	66.04	49.33	62.50	**99.60**
Depth,RGB,PCD (MVCNN,MVCNN,PointNet)	**87.99**	35.72	**49.68**	**75.31**	**60.49**	**77.45**	**73.79**	**68.44**	96.72
Depth,RGB,PCD (MVCNN,GQN,PointNet)	86.07	**35.97**	47.23	73.10	60.46	74.30	63.08	67.83	99.02

**Table 4 sensors-20-04761-t004:** Results of change caption correctness evaluation using s15_o10 (top 11 rows), s15_o85 (middle 6 rows), and s100_o10 (bottom 6 rows). The highest scores are shown in bold.

Modality (Encoder)	Accuracy (%)			
	Change Type	Object	Color	Class
Depth (MVCNN)	44.54	26.82	35.11	36.78
RGB (MVCNN)	49.26	49.52	62.16	53.18
RGB (GQN)	74.48	69.41	79.54	71.88
PCD (PointNet)	**97.24**	53.65	59.20	65.44
Depth,RGB (MVCNN,MVCNN)	73.59	62.53	73.74	66.38
Depth,RGB (MVCNN,GQN)	78.44	72.46	81.84	74.91
Depth,PCD (MVCNN,PointNet)	93.03	39.64	48.75	52.06
RGB,PCD (MVCNN,PointNet)	93.37	58.52	70.68	65.29
RGB,PCD (GQN,PointNet)	93.94	74.39	83.85	77.39
Depth,RGB,PCD (MVCNN,MVCNN,PointNet)	93.09	63.02	73.32	69.32
Depth,RGB,PCD (MVCNN,GQN,PointNet)	94.96	**75.35**	**84.05**	**78.30**
Depth (MVCNN)	52.42	9.13	22.10	19.73
RGB (MVCNN)	46.38	17.23	38.99	23.34
RGB (GQN)	59.68	17.33	34.45	26.77
PCD (PointNet)	**96.17**	14.62	27.53	27.28
Depth,RGB,PCD (MVCNN,MVCNN,PointNet)	92.55	**23.51**	**44.92**	**32.18**
Depth,RGB,PCD (MVCNN,GQN,PointNet)	90.75	20.29	37.65	**32.18**
Depth (MVCNN)	42.88	19.85	29.70	31.29
RGB (MVCNN)	50.66	43.58	58.18	47.73
RGB (GQN)	52.00	40.28	52.13	45.72
PCD (PointNet)	90.49	33.73	44.60	47.13
Depth,RGB,PCD (MVCNN,MVCNN,PointNet)	**91.54**	**55.28**	**69.60**	**61.64**
Depth,RGB,PCD (MVCNN,GQN,PointNet)	90.07	48.20	59.93	55.77

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
