# Peer review of "Indoor Scene Change Captioning Based on Multimodality Data"

_sensors, 2020, doi:10.3390/s20174761_

Round 1

Reviewer 1 Report

The main concern regarding the work is its applicability. It would require a set of multiple sync RGBD cameras in the room to work properly. This issue needs more elaboration, since there could be interferences among the RGBD cameras due to the projected patterns, or other problems, like the degree of sync…. that will hinder or avoid its real applicability, especially since the current experiment is carried out based on virtual datasets. 

Related with the previous, the present workflow relies not only on the use of RGBD cameras, but in a room’s point cloud as an additional input (called PCD by the authors). Since there are available multiple sync range images, why did not the authors fuse them to generate the point cloud? 

Besides, authors should state clearly what they call PCD, since in the line 268 the sentence “we transformed the mesh data to PCD” implies that were extracted the triangle vertexes to a PCD file. A point cloud generated from interpolation of a mesh would be different than a point cloud created as a list of mesh triangles’ vertexes.

Figure 1 caption should state clearly that the multiple RGB and range images are from multiple viewpoints of the same scene acquired simultaneously. 

Figures 2 and 3 are shown before their citation in text.

In line 263, the authors stated that they sampled the cuboid rooms. If the term sample is referred to the spatial sampling, the authors should stated the sampling resolution, in order to evaluate the applicability with other RGBD cameras.

Line 265: “we arranged the selected objects at random navigable positions in the room scene”. Please elaborate more. How and by what software was carried out the process of arranging objects?

Author Response

First of all, we thank the reviewers for the constructive comments. We did our best to revise the manuscript, including the parts that they pointed out. In our revised manuscript, we highlight the parts modified regarding comments from reviewers in purple. Please see the attachment for the details of our responses to the comments.

Reviewer 2 Report

The article describes a novel architecture of the deep learned neural system for scene changes captioning. Advantages of the article: 1) The novel system was implemented using cloud computing. 2) A large artificial collection of labeled scenes captured by cameras, lidars and cloud point appliances was prepared. 3) The system was tested and measured. 4) A good literature review is given. Disadvantages of the article: 1) It is hard for the reader to recreate the author's system and repeat experiments. 2) The system is not compared to any other known from the literature.

Author Response

(The authors gave the same response as above.)

Reviewer 3 Report

The article deals with the indoor scene change captioning by a combination of analysis and synthesis methods. The analysis consists in the decomposition of 3 input data of a given image (RGB, Depth and point cloud data - PCD) into an encoded sequence of elements. The essence of the synthesis is to compare the results of the sequence before and after change observation. The result then goes to the caption generator block. In my opinion, this is an innovative method in the field.

I have a comment on the conclusion regarding the amount of work, which seems to be relatively brief. The authors should compare the proposed approach (state the advantages and disadvantages) with the solution of similar methods of indoor scene captioning in the related literature.

Author Response

(The authors gave the same response as above.)

Round 2

Reviewer 1 Report

Thank you for updating the paper according to the review issues and the answers provided. I consider that the manuscript is suitable for publication.